# *Ceratocarpus arenarius*: Botanical Characteristics, Proximate, Mineral Composition, and Cytotoxic Activity

**DOI:** 10.3390/molecules29020384

**Published:** 2024-01-12

**Authors:** Aigerim Kantureyeva, Gulbaram Ustenova, Alenka Zvonar Pobirk, Serzhan Mombekov, Moldir Koilybayeva, Akerke Amirkhanova, Nadezhda Gemejiyeva, Assem Mamurova, Nina Kočevar Glavač

**Affiliations:** 1School of Pharmacy, S.D. Asfendiyarov Kazakh National Medical University, Tole-bi 94, Almaty 050012, Kazakhstan; ustenova.g@kaznmu.kz (G.U.); mombekov.s@kaznmu.kz (S.M.); koilybayeva.m@kaznmu.kz (M.K.); amirhanova.a@kaznmu.kz (A.A.); 2Department of Pharmaceutical Technology, Faculty of Pharmacy, University of Ljuljana, Aškerčeva cesta 7, 1000 Ljubljana, Slovenia; alenka.zvonar-pobirk@ffa.uni-lj.si; 3Laboratory of Plant Resources, Institute of Botany and Phyto-Introductions, Almaty 050000, Kazakhstan; ngemed58@mail.ru; 4Department of Biodiversity of Bioresources, al-Farabi Kazakh National University, Almaty 050040, Kazakhstan; amamurova81@mail.ru; 5Department of Pharmaceutical Biology, Faculty of Pharmacy, University of Ljubljana, Aškerčeva cesta 7, 1000 Ljubljana, Slovenia

**Keywords:** anatomy, *Ceratocarpus arenarius*, Chenopodiaceae, cytotoxic activity, mineral content, proximate composition

## Abstract

*Ceratocarpus arenarius* (Chenopodiaceae) is an under-investigated annual plant that occurs in dry areas stretching from eastern and south-eastern Europe to East Asia. This article presents the botanical characterization and examination of proximate parameters, minerals and cytotoxic activity of *C. arenarius* that grows wild in Kazakhstan. The results of morphological analysis using a light microscope, based on cross-sections of stems, roots and leaves, provide the necessary data to develop a regulatory document for this herbal substance as a raw material for use in the pharmaceutical, cosmetic and food industries. The investigated proximate characteristics included moisture content (6.8 ± 0.28%), ash (5.9 ± 0.40%), fat (12.5 ± 21.28%) and protein (392.85 ± 25.50). The plant is also rich in minerals (mg/100 g dry weight); Na (20.48 ± 0.29), K (302.73 ± 1.15), Zn (4.45 ± 0.35), Fe (1.18 ± 0.03), Cu (0.11 ± 0.02), Mn (0.76 ± 0.01), Ca (131.23 ± 0.09) and Mg (60.69 ± 0.72). The ethanolic extract of *C. arenarius* showed no acute toxicity against the brine shrimp nauplii.

## 1. Introduction

Chenopodiaceae, in the order Caryophyllales, is one of the most interesting families with respect to having species with a large diversity in the structure of carbon-absorbing organs responsible for different types of photosynthesis [1]. The Chenopodiaceae family includes annual or perennial herbs, subshrubs, shrubs, small trees and climbers. They are found in deserts, semi-deserts, salt-marshes, and coastal or inland saline and ruderal sites [2]. The family includes about 98 genera and 1400 species distributed mainly in the extra tropical regions of the Northern Hemisphere [3]. 

Plant species belonging to the Chenopodiaceae family show high drought and salinity resistance and tolerance to nutrient deficiency and often grow in psammophytic or halophytic plant communities. They are considered pioneer plants in the colonization and settlement of harsh edaphic environments affected by salt or drought, and therefore, they play crucial roles in erosion control and rehabilitation of desert ecosystems [4]. 

The large territory of arid and semi-arid zones and diverse soil and climatic conditions, including extreme environmental conditions, are the reason for the rich genetic resources of Kazakhstan, Central Asia. About 6000 species of vascular plants from 160 families are distributed in the country [5,6].

*Ceratocarpus arenarius* L. (Figure 1) is an annual herbaceous plant belonging to the family of Chenopodiaceae. It is native to Kazakhstan (Figure 2), as well as other areas stretching from eastern and south-eastern Europe to East Asia, and is locally known as tumbleweed. It is usually found in dry climates with precipitation of 100–400 mm, in deserts, arid slopes, sands, wastelands and along roadsides. Plants produce fruits (utricles) near the soil surface (basicarps) and a continuous series of morphologically distinct fruits from the lower to upper parts of the canopy [7]. The seeds are numerous, approx. 4000 per plant, with a thick pericarp that fits tightly to the seed coat. In autumn, which is the end of the growing season, the above-ground parts of *C. arenarius* are easily separated from the soil surface and dispersed long distances by the wind [8].

*C. arenarius* is widely used in folk medicine and agriculture. It is a potential source of biologically active compounds such as polyphenolic compounds, e.g., flavonoids and phenolic acids, organic acids, saponins, steroids, and vitamins C and B2 [9,10]. To support the possibility of using this plant in medicine and the pharmaceutical, cosmetic and food industries, it is necessary to develop a regulatory document or a monograph for the herbal substance as a raw material. Important sections of such monographs, which determine the authenticity of plant raw materials, and ensure reliable quality control thereof, are morphological characteristics, both at a macroscopic and microscopic level. 

Such characterization has not yet been available for *C. arenarius*. Therefore, the purpose of this work was to conduct a detailed morphological and anatomical study of *C. arenarius* that grows wild in Kazakhstan, as well as the plant’s nutrient and mineral contents and cytotoxic activity.

## 2. Results and Discussion

### 2.1. Morphological Characteristics

The *C. arenarius* plant (Figure 1) is grayish-green in color, and its plant height reaches 9.5 ± 1.16 cm. The plant is strongly and repeatedly fork-branched from the base, and the leaves often form a spherical shape. The leaves are alternate, narrowed to the base, single-nerved, with stellate hairs. They are aromatic in nature upon bruising/crushing in hands. The average leaf length is 3.5 ± 0.07 cm, and the number of leaves per plant is 37.2 ± 1.1. The stem is angular in shape and dark green in color. The average stem length is 11.4 ± 0.76 cm. The average seed length is 6.27 ± 0.12 mm. Morphological features of *C. arenarius* stems, leaves and seeds are presented in Table 1 and Figure 3.

### 2.2. Anatomical Characteristics

The anatomical characteristics of *C. arenarius* were assessed for the leaf, stem and root. The recorded observations are described below.

#### 2.2.1. Microscopic Structure of the Leaf

The leaf blade is of dorsiventral type, and upper and lower epidermises were observed in detail, which are in the leaf structure (Figure 4). The cells of the epidermis of the upper and lower sides of the leaf have irregularly thickened walls and become slightly elongated along the length of the leaf lobe. The mesophyll has the palisade and the spongy parenchyma composed of more compact and smaller vascular bundles, as compared to other leaf tissues. Under the upper epidermis, the palisade mesophyll is organized into two rows. The spongy mesophyll is barely noticeable. Well-developed trichomes are numerous, 6 to 7.

The leaf trichomes of the stellate hairs have a one- or two-cell stalk followed by more densely packed cells, each of which is elongated on one or both sides, forming together a star-shaped cap. In the leaves stained with safranin and astra-blue, the walls of these trichomes were colored red to varying degrees and stood out on the blue-colored cells of the epidermis. After treatment with the phloroglucin/HCl reagent, the trichomes were not stained, which demonstrates the absence of lignification of cell walls. Stellate hairs were insensitive to 10% potassium hydroxide and retained their structure. 

Biometric indicators of the leaf structure of *C. arenarius* are presented in Table 2. The thickness of the leaf is 1.99 ± 0.18 µm. The cells of the lower epidermis (0.25 ± 0.05 µm) are larger and arranged in 1–2 rows, and the thickness of the palisade mesophyll (1.98 ± 0.02 µm) is expressed better than the thickness of the spongy mesophyll (0.49 ± 0.13 µm). The vascular bundle (0.148 ± 0.13 µm) is collateral, either single or in a group of two, arranged at the ridges. 

There are numerous stomata on the lower surface of the leaf. The neighboring epidermal cells of the stoma show a rectangular shape and a chain-like appearance. The stoma consists of a pair of specialized cells called guard cells, which regulate the degree of openness of the stoma, and between them is the stomata gap. The walls of the guard cells are thickened unevenly: those directed to the gap are thicker than the walls directed from the gap. The gap widens and narrows, regulating transpiration and gas exchange. Companion cells are radially arranged around the stomatal cells, forming an actinocytic stomata type (Figure 5).

#### 2.2.2. Microscopic Structure of the Stem 

The stem is covered with an epidermis followed by the primary cortex, the central cylinder and the core (Figure 6). The main tissues of the primary cortex are represented by the lamellar collenchyma, which is located either in a solid ring or in separate sections, and vascular tissue, i.e., phloem and xylem. The stem is also characterized by the presence of lignified sclerenchyma cells. The central cylinder is formed from the sclerenchyma and medullary parenchyma of pericyclic sagging adjacent to the starch-bearing vagina. The other part of it is filled with parenchyma in which conducting bundles are arranged in one circle. The main biometric indicators of the stem include the thickness of the epidermis, thickness of the primary parenchyma, thickness of the collenchyma, the diameter of xylem and phloem, thickness of conducting beam and the diameter of core parenchyma zone (Table 3).

#### 2.2.3. Microscopic Structure of the Root 

Sections were made from fresh samples using a freezing microtome; no dyes were used. In the cross-section (Figure 7), the roots are thin, cylindrical, light brown and have a well-developed plug, cortex, phloem and a central core of primary and secondary xylem. In the root cylindrical structure, there is the outermost exoderm layer. The periderm layer (0.16 ± 0.02 µm) is located under the exoderm layer. The epidermis is narrow. The conductive bundles form small areas separated by broad medullary rays consisting of tangentially elongated cells. The xylem is developed and well-visible at 0.24 ± 0.01 µm in diameter. Secondary phloem (0.32 ± 0.01 µm in diameter) is clearly differentiable in loose sieve elements and elongated horizontally and arranged in phloem rays. The location of the phloem and xylem is radial and scattered. The conducting bundles are arranged in an order close to the central circle. The cell thickness of the primary cortex (0.48 ± 0.03 µm) is densely well developed. Biometric measurements of the root are shown in Table 4 below.

### 2.3. Determination of Proximate Parameters

The proximate composition of *C. arenarius* is presented in Table 5.

Ash content and moisture content were 6.8% and 5.9%, respectively. High ash content indicates significant concentrations of several minerals. Ashing eliminates all of the sample’s organic constituents; ash also contains inorganic plant parts. Ash indicates that a plant is highly digestible [11]. Depending on the environment and the physiology of the plant, different plant species have different moisture contents. Moisture content is influenced by environmental factors such as temperature, humidity, weather, harvest time and storage conditions [12], and a moisture content of more than 15% is not desirable because of the risk of microbiological contamination. The moisture content of 6.8% found in the tested plant material corresponds to a typical moisture content of 6–8% [13].

The fatty-acid profile of the *C. arenarius* plant is presented in Table 6 and is expressed as a percentage (%) of total fatty acids (TFA). The profiling of fatty acids showed the presence of eight compounds; of these, four were saturated fatty acids (SFAs), two were monounsaturated fatty acids (MUFAs), and two were polyunsaturated fatty acids (PUFA). Fatty acids containing one or more covalent double bonds between carbon–carbon at various positions on the carbon chain are called unsaturated fatty acids. In general, a high percentage of unsaturated fatty acids (83.8%) was found. In addition, percentages of MUFAs (62.5%) were higher than those of PUFAs (21.3%). The major unsaturated fatty acids were oleic (62.2%) followed by linoleic (20.8%), linolenic (0.5%) and palmitoleic (2.29%) acids. Oleic acid is considered beneficial, as it has been shown to lower cholesterol levels in low-density lipoproteins [14]. Polyunsaturated fatty acids (PUFAs) are basic components involved in the architecture and function of cellular membranes and play key roles in several biological processes. Linoleic acid is a major constituent of human tissues, and it is considered to be an essential fatty acid [15]. Fatty acids that consist of a single covalent bond between carbon–carbon atoms and are generally solid at room temperature are called saturated fatty acids. Palmitic acid (9.3%), stearic acid (5.6%), myristic acid (0.8%) and pentadecanoic acid (0.5%) found in vegetable oils are the most important saturated fatty acids. Saturated fatty acids can be synthesized in the human body, and even if no fat is consumed, these types of fatty acids can be synthesized from molecules formed by carbohydrate metabolism. In response to this, unsaturated fats are essential fatty acids that the body needs. They are liquid at room temperature and most of them are of vegetable origin [16,17].

The fatty acid composition is a good indicator of the quality and stability of the oil. Consequently, it is vital to determine the type and content of fatty acids [18]. The season of collection and other abiotic factors including soil, salinity, light and temperature may also contribute to variations in fatty acid content [19,20]. The current investigation represents the first study to assess the fatty-acid composition in *C. arenarius*. 

Amino acids play a number of roles in the body and are required for the synthesis of proteins. It is necessary to have a balanced diet that contains all of the amino acids—the non-essential ones and those that are not synthesized in the body (essential amino acids). *C. arenarius* contains essential and non-essential amino acids. Table 7 shows that amino acids with the highest amounts detected include glutamic acid (2298 mg/100 g dry weight), aspartic acid (1204 mg/100 g weight) and alanine (680 mg/ 100 g dry weight). This is the first study that reports the amino acid composition of *C. arenarius*.

The analysis of the mineral composition is shown in Table 8. The findings revealed that *C. arenarius* contained eight important minerals. The macro minerals analyzed included sodium (Na), potassium (K) and calcium (Ca). The micro minerals included magnesium (Mg), copper (Cu), zinc (Zn), iron (Fe) and manganese (Mn). K and Ca had the highest mineral concentrations of 302.725 and 131.230 mg/100 g of dry weight, respectively. The lowest concentration minerals were Cu, Mn and Fe, with respective concentrations of 0.11, 0.76 and 1.18 mg/100 g of dry weight. 

For the optimal functioning of human systems as well as for healthy development and growth, mineral elements are needed in trace amounts [21]. The most prevalent mineral in bones is Ca, which is necessary for many cellular processes such as neuron and muscle function, hormone responses and, blood clotting [22]. The ionic equilibrium of the human body and tissue excitability are maintained by Na and K. Na is crucial for the transfer of metabolites, and K is crucial due to its diuretic properties. Any food’s K/Na ratio is a crucial element linked to hypertension and atherosclerosis. Na raises blood pressure, while K lowers it [23]. Magnesium helps to avoid immune system dysfunction, impaired spermatogenesis, impaired muscle development, growth retardation, cardiomyopathy and bleeding disorders [24]. The stabilization of macromolecular production and structure is facilitated by Zn. Both DNA and RNA polymerases are Zn-dependent enzymes, and the role of the metal ion in DNA and RNA synthesis is well known [25]. The cofactor for enzymes such as arginase and glycosyl transferase is manganese (Mn). Other enzymes, such as glutamine synthetase and phosphoenol pyruvate carboxy kinase, are also triggered by Mn ions. Mn is necessary for the synthesis of hemoglobin as well [26]. Fe performs a wide range of biological functions, including the role in hemoglobin and the transfer of oxygen from the lungs to tissue cells [27]. Fe deficiency affects humans more frequently than any other nutritional deficiency [28]. The Cu protein is made up in a large part by Cu. The three main Cu-containing metalloenzymes are tyrosine oxidase, lysyl oxidase and cytochrome C oxidase.

As part of enzymatic defense, microelements such as Zn, Mn, Cu and Fe are crucial in the fight against oxidative stress. Some plants contain significant amounts of minerals; their presence and abundance are influenced by the plant’s ancestry, history and phytochemical traits [29].

### 2.4. Cytotoxic Activity

*Artemia salina* is a simple biological organism (marine invertebrate) about 1 mm in size. Their freeze-dried cysts (*A. salina* eggs) can last for several years and can be hatched into larvae without special equipment. The brine shrimp lethality test (BSLT) or brine shrimp survival method is a general bioassay that has been used successfully for preliminary assessment of cytotoxicity [30,31]. This bioassay can be employed to determine the cytotoxic activity of plant extracts, and it is a very useful tool for screening a wide range of chemical compounds [32]. 

The results of cytotoxic activity are shown in Table 9. The comparison drug, actinomycin D, showed cytotoxicity at all concentrations—the mortality of nauplii was 63–96%. In contrast, the ethanolic extract of *C. arenarius* showed no toxicity against *A. salina*. Even with the increase in the extract concentration, no mortality was observed.

## 3. Materials and Methods

### 3.1. Plant Material

The plant of *C. arenarius* (Figure 8) was collected in the Almaty region (Kazakhstan) in June 2021. The material was identified at the Institute of Botany and Phyto introduction of the Republic of Kazakhstan, and a voucher specimen was registered and stored under No. 01-09/305. The raw material was crushed in an electric mill (Figure 8) and then preserved in clean boxes.

### 3.2. Morphological Studies

The macroscopic study of vegetative organs was carried out in accordance with the requirements of the State Pharmacopoeia of the Republic of Kazakhstan [33]. Morphological measurements were taken on roots, stems, leaves and seeds, using 10 individual plants. Values were statistically processed using Microsoft Excel 2016, and final results were expressed as a mean value ± SD.

#### Anatomical Studies

Anatomical studies of vegetative organs were conducted in accordance with the State Pharmacopoeia of the Republic of Kazakhstan and methods described by Vekhov et al. [34]. Dry raw material (whole plants) was soaked in a mixture of glycerin:water:ethanol 96% (1:1:1). Anatomical preparations were prepared using a freezing microtome (Minux S700, Shenzhen, China) [35]. The thickness of anatomical sections varied between 10 and 15 μm. Microphotographs of anatomical sections were taken using an MC-300 microscope (Micros Company, Vienna, Austria) magnification 10× for ocular and 40× for objective) with a CAMV400/1.3 m video camera (Micros Company, Vienna, Austria). Mathematical processing of the obtained images was carried out according to the methodology described in [36]. Cross sections of the leaf, stem and root were prepared. Clarification of the preparations was carried out with glycerin. More than 100 temporary preparations were prepared. Each parameter for an individual preparation was measured tenfold.

### 3.3. Determination of Proximate Parameters

#### 3.3.1. Determination of Moisture

The method outlined by AOAC (2016) [37] was used to determine moisture content. A desiccator was used to allow an empty crucible to cool and dry to a consistent weight before being weighed (W1). Dry plant sample (2.0 g) was dried at 105 °C until it reached a constant weight, then it was weighed (W2) in the crucible. The plant sample-containing crucible was allowed to cool in a desiccator before the weight (W3) was calculated. The percentage used to calculate moisture content was as follows:$$moisture\left(\%\right)=\frac{\left(W1-W2\right)\times 100}{W1}$$
where
W1 = weight (g) of sample before drying;W2 = weight (g) of sample after drying.

#### 3.3.2. Determination of Ash

The ash content assay was conducted using the AOAC (2016) [37] method. In order to determine the dry weight (W1), a heat-resistant porcelain crucible was dried for 10 min at 105 °C in an oven before cooling in a desiccator. Then, 2.0 g of the ground-up plant sample was weighed again (W2) after being measured in the porcelain crucible. To guarantee appropriate ashing, the crucible containing the sample was burned in a furnace at 250 °C for 1 h and then at 550 °C for 7 h. After being taken out, the crucible was weighed (W3) after cooling in a desiccator. Ash content percentage was calculated as follows:$$ashcontent\left(\%\right)=\frac{\left(W3-W1\right)\times 100}{W2-W1}$$
where
W1 = weight of empty crucible;W2 = weight of crucible + sample;W3 = weight of crucible + ash.

#### 3.3.3. Determination of Fatty Acid Composition

One volume of sample was extracted by 20 times the volume of chloroform:methanol (2:1) for 5 min. Then, the mixture was filtered through a paper filter to obtain a clear extract that was evaporated in a round bottom flask on a rotary evaporator at a bath temperature of 30–40 °C until dried. Then, 10 mL of methanol and 2–3 drops of acetyl chloride were added, and methylation reaction was performed at 60–70 °C for 30 min. Then, methanol was evaporated on a rotary evaporator, and the dry residue was dissolved in 5 mL of hexane. An aliquot of the upper hexane layer was directly taken and analyzed using GC-MS system (Carlo Erba 4200, Cornaredo, Italy) with capillary column (30 m × 0.25 mm, 0.25 µm). Helium is used as a carrier gas. It was operated under the following conditions: oven temperature of 188 °C for 1 h. The injector temperature was set at 188 °C, and detector temperature was set at 230 °C [38].

The identification of the compounds was based on a comparison of their mass spectra and retention indices with those of the synthetic compounds spectral library of the National Institute of Standards and Technology (NIST11). 

#### 3.3.4. Determination of Amino Acid Composition

In total, 1 g of the sample was hydrolyzed in 5 mL of 6 N hydrochloric acid at 105 °C for 24 h, in ampoules sealed under a jet of argon. The resulting hydrolysate was dried on a rotary evaporator three times at a temperature of 40–50 °C. The dried mass was dissolved in 5 mL of sulfosalicylic acid. The supernatant was passed at a rate of 1 drop/second through a column of Daux 50 ion-exchange resin after being centrifuged for 5 min. The resin was then rinsed until the pH was neutral. To elute amino acids from the column, 3 mL of 6 N NH_4_OH solution was passed through the column at a rate of 2 drops/second. The eluate was collected in a round bottom flask together with demineralized water, which was used to wash the column to neutral pH. Then, the contents of the flask were evaporated to dryness on a rotary evaporator at a pressure of 1 atm and temperature of 40–50 °C. In total, 1 drop of freshly prepared 1.5% SnCl_2_ solution, 1 drop of 2,2-dimethoxypropane and 1–2 mL of propanol saturated with hydrochloric acid were added, and the mixture was heated to 110 °C, kept at this temperature for 20 min, and then, the contents were evaporated from the flask again on a rotary evaporator. In the next step, 1 mL of freshly prepared acetylating reagent (acetic anhydride:triethylamine:acetone = 1:2:5, *v*/*v*) was introduced into the flask and heated at 60 °C for 1.5–2 min. Then, the sample was evaporated again on the rotary evaporator to dryness, and 2 mL of ethyl acetate and 1 mL of saturated NaCl solution were added to the flask. The contents were thoroughly mixed, and as two layers of liquids were clearly formed, the upper one (ethyl acetate) was taken for gas chromatographic analysis, which was carried out on a gas-liquid chromatograph (Carlo Erba Reagents, Cornaredo, Italy) [38].

#### 3.3.5. Determination of Mineral Content

The procedure was modified from that given by Idris et al. [39]. The plant samples that had been ground up (3 g) were weighed and ashed at 550 °C. The resultant residue, known as white ash, was dissolved in 4 mL of concentrated HCl, filtered, and the filtrate was then diluted with distilled water in a volumetric flask. The examination of minerals was then performed on the extract’s final solution. Nutrient analysis of K, Ca, Mg, Fe, Mn, Zn, Na and Cu contents were determined using Atomic Absorption Spectrophotometer (Analytik Jena nova 350, Jena, Germany). The analyses were performed in triplicate.

### 3.4. Extraction

The freshly cleansed plant was air-dried for 72 h at room temperature before being ground in an electric blender. The powder (100 g) was extracted for 48 h with 1 L of 70% ethanol in a glass conical flask using a shaker at 25 °C and filtered. Using a rotary evaporator under low pressure, the crude extract was concentrated at 40 °C. The extract was preserved after drying at 4 °C until further analysis.

### 3.5. Cytotoxic Activity

*A. salina* was taken to determine cytotoxic activity. This technique is based on determining the difference between the numbers of dead *A. salina* larvae in the sample analyzed (experiment) and water that does not contain toxic substances (control). The criterion for acute lethal toxicity of a substance solution is the death of 50% or more larvae in the experiment compared to the control. The flask was filled with artificial seawater, and *A. salina* eggs were added. They were kept for 72 h with a soft air supply until the shrimp hatched from the eggs. Actinomycin D was used as a comparison drug. Samples (70% ethanol extracts of *C. arenarius*) were tested at concentrations of 1, 5 and 10 mg/mL. Each sample was tested in three parallel experiments conducted at a temperature of 20 °C, in natural light period. The salinity of the control artificial water was 8.0–8.5 (pH). At the time of the biotest, the *A. salina* larvae were up to 1 day old. The planting density of larvae was 20–40 specimens per test tube. 

## 4. Conclusions

This study provided analysis of morpho-anatomical characteristics, nutritional contents and cytotoxic activity of wild-growing *C. arenarius*. The results of botanical characterization have an important diagnostic significance and will allow the evaluation of authenticity of *C. arenarius* as a plant raw material in the development of regulatory documentation. This will expand the raw material base for obtaining new phytopreparations for therapeutic, cosmetic or nutritional use. In addition, our studies of the plant’s nutrient and mineral contents and cytotoxic activity confirm its promising potential. However, as a largely unexplored plant, *C. arenarius* should first be subjected to detailed phytochemical characterization in further studies.

## Figures and Tables

**Figure 1 molecules-29-00384-f001:**
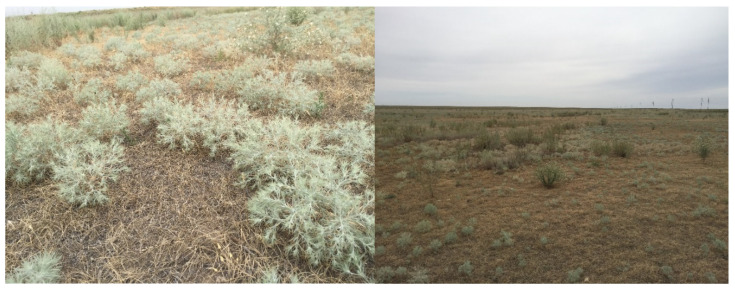
*C. arenarius*, Almaty region, Kazakhstan.

**Figure 2 molecules-29-00384-f002:**
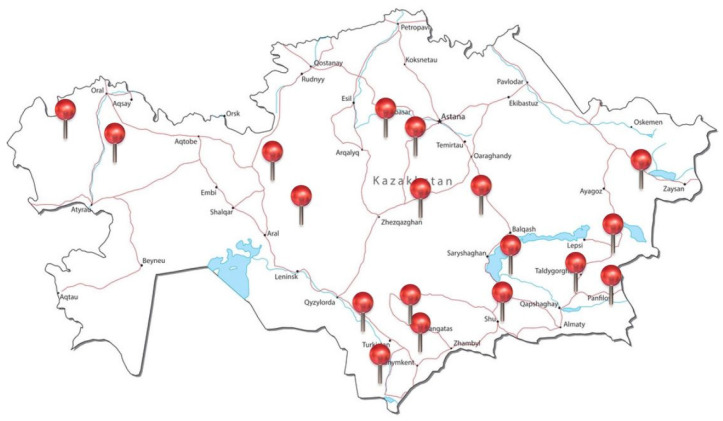
Distribution of *C. arenarius* over Kazakhstan. The location where the plant is widespread is indicated in red.

**Figure 3 molecules-29-00384-f003:**
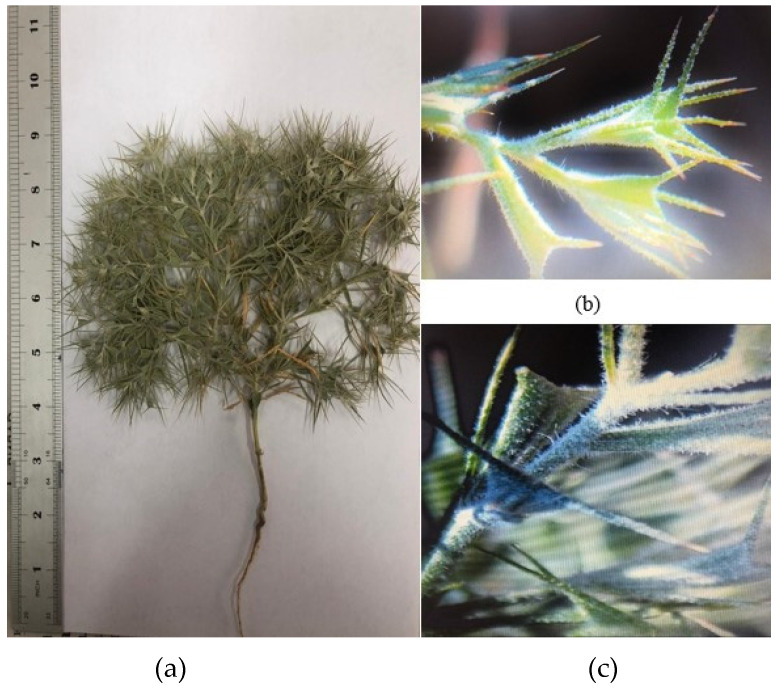
Raw material sample of *C. arenarius*. (**a**) Whole plant. (**b**) Leaf. (**c**) Stem.

**Figure 4 molecules-29-00384-f004:**
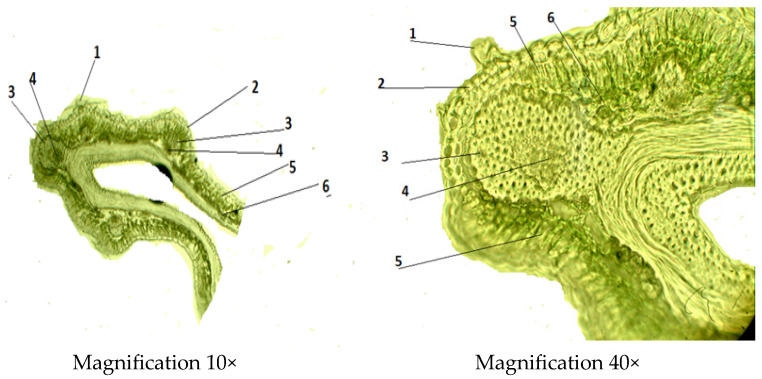
Cross section of the *C. arenarius* leaf; 1—trichome; 2—epidermis; 3—xylem; 4—phloem; 5—palisade mesophyll; 6—spongy mesophyll.

**Figure 5 molecules-29-00384-f005:**
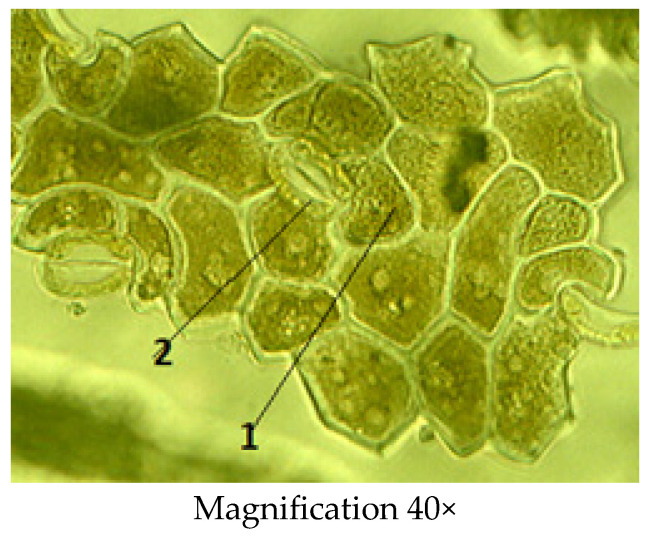
The lower epidermis of the *C. arenarius* leaf; 1—stomatal gap; 2—guard cells (actinocytic).

**Figure 6 molecules-29-00384-f006:**
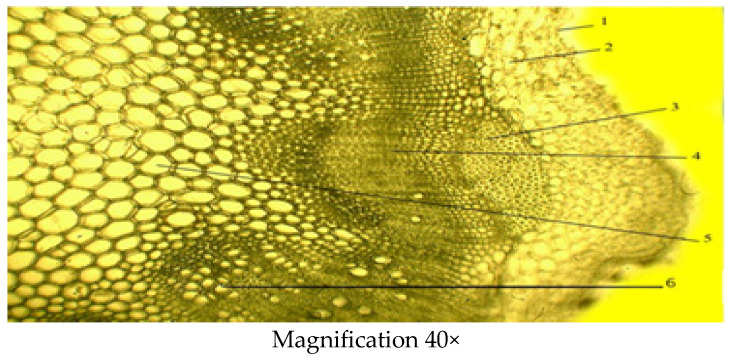
Cross section of the *C. arenarius* stem; 1—epidermis; 2—lamellar collenchyma; 3—pericyclic sclerenchyma; 4—phloem; 5—medullary parenchyma; 6—xylem.

**Figure 7 molecules-29-00384-f007:**
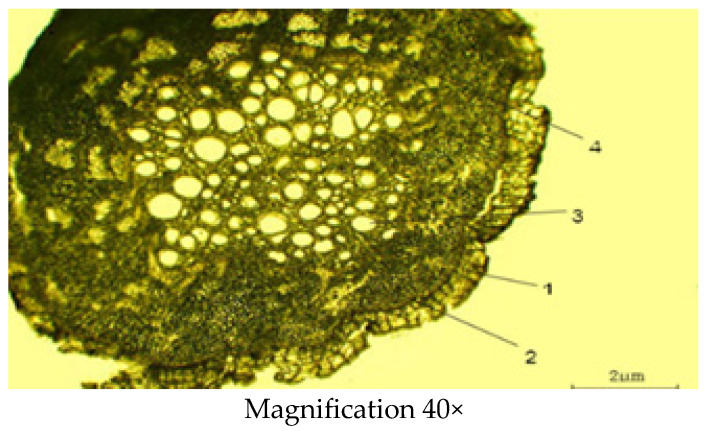
Cross section of the *C. arenarius* root; 1—periderm; 2—periderm; 3—xylem; 4—phloem.

**Figure 8 molecules-29-00384-f008:**
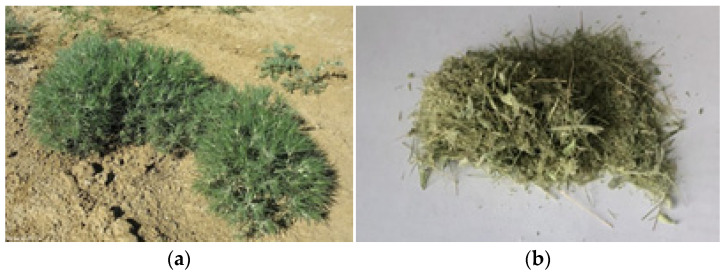
The plant of *C. arenarius* (**a**); powdered plant (**b**).

**Table 1 molecules-29-00384-t001:** Macroscopic characterization of *C. arenarius*. Values are represented as a mean value ± standard deviation.

Sample	Indicator/Mean ± SD
Herb color	grayish-green
Odor	Aromatic
Plant height (cm)	9.5 ± 1.16
Phyllotaxis	Alternate
Flowers	single, pale yellow
Leaf length (cm)	3.5 ± 0.07
Number of leaves per plant	37.2 ± 1.1
Stem length (cm)	11.4 ± 0.76
Seed length (mm)	6.27 ± 0.12

**Table 2 molecules-29-00384-t002:** Biometric measurements of the anatomical structures of *C. arenarius* leaf. Values are represented as a mean value ± standard deviation.

Indicator	Mean ± SD (µm)
Thickness of the leaf	1.99 ± 0.18
Thickness of the upper epidermis	0.03 ± 0.02
Thickness of the lower epidermis	0.25 ± 0.05
Thickness of the spongy mesophyll	0.49 ± 0.13
Thickness of the palisade mesophyll	1.98 ± 0.02
Diameter of the vascular bundle	0.148 ± 0.13

**Table 3 molecules-29-00384-t003:** Biometric measurements of the anatomical structures of *C. arenarius* stem. Values are represented as a mean value ± standard deviation (*n* = 10).

Indicator	Mean ± SD (µm)
Thickness of the epidermis	0.02 ± 0.003
Thickness of the primary parenchyma	0.27 ± 0.02
Thickness of collenchyma	0.38 ± 0.14
Diameter of xylem	0.22 ± 0.04
Diameter of phloem	0.38 ± 0.09
Thickness of conducting beam	1.06 ± 0.01
Diameter of core parenchyma zone	3.27 ± 0.003

**Table 4 molecules-29-00384-t004:** Biometric measurements of the anatomical structures of *C. arenarius* root. Values are represented as a mean value ± standard deviation.

Indicator	Mean ± SD (µm)
Thickness of the periderm	0.16 ± 0.02
Thickness of the primary cortex or cortex parenchyma	0.48 ± 0.03
Diameter of the central cylinder	2.89 ± 0.25
Diameter of xylem	0.24 ± 0.01
Diameter of phloem	0.32 ± 0.01

**Table 5 molecules-29-00384-t005:** Proximate composition of *C. arenarius*.

Parameter	Value
Moisture (%)	6.8 ± 0.28
Protein (mg/100 g)	392.85 ± 25.50
Ash (%)	5.9 ± 0.40
Fat (%)	12.5 ± 21.28

**Table 6 molecules-29-00384-t006:** Fatty acid profile of *C. arenarius*.

№.	Parameter	C Number: Number of Double Bonds	Class of Compound	Content, %
1	Myristic acid	14:0	Saturated	0.8
2	Pentadecanoic acid	15:0	Saturated	0.5
3	Palmitic acid	16:0	Saturated	9.3
4	Palmitoleic acid	16:1	Monounsaturated	0.3
5	Stearic acid	18:0	Saturated	5.6
6	Oleic acid	18:1	Monounsaturated	62.2
7	Linoleic acid	18:2	Polyunsaturated	20.8
8	Linolenic acid	18:3	Polyunsaturated	0.5

**Table 7 molecules-29-00384-t007:** Amino acid composition of *C. arenarius*.

Amino Acid	Content mg/100 g
Alanine	680
Glycine	222
Leucine	374
Isoleucine	345
Valine	236
Glutamic acid	2298
Threonine	218
Proline	435
Methionine	56
Serine	370
Aspartic acid	1204
Cysteine	28
Oxyproline	1
Phenylalanine	352
Tyrosine	384
Histidine	183
Ornithine	1
Arginine	256
Lysine	16 2
Tryptophan	52

**Table 8 molecules-29-00384-t008:** Mineral content of *C. arenarius*. Results are represented as a mean value± standard deviation.

Mineral (mg/100 g Dry Weight)							
K	Ca	Mg	Fe	Na	Mn	Zn	Cu
302.73 ± 1.15	131.23 ± 0.09	60.69 ± 0.72	1.18 ± 0.03	20.48 ± 0.29	0.76 ± 0.01	4.45 ± 0.35	0.11 ± 0.02

**Table 9 molecules-29-00384-t009:** Cytotoxic activity of the 70% ethanolic extract of *C. arenarius*.

Sample	Concentration mg/mL	% of Surviving Nauplii in the Control	% of Surviving Larvae in the Sample	% Mortality
Actinomycin D	10	96	0	96
	5	96	4	92
	1	96	33	63
Extract	10	96	96	0
	5	96	96	0
	1	96	96	0

## Data Availability

The authors confirm that the data supporting the findings of this study are available within the article and Appendix A.

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
