# Peer review of "Ceratocarpus arenarius: Botanical Characteristics, Proximate, Mineral Composition, and Cytotoxic Activity"

_molecules, 2024, doi:10.3390/molecules29020384_

Round 1

Reviewer 1 Report

Comments and Suggestions for Authors

Authors have presented the Pharmaceutical, cosmetic and food potential of Ceratocarpus arenarius: Botanical characteristics, proximate and mineral composition, and cytotoxic activity. However, the manuscript does not contain any experiments and results related to Pharmaceutical and cosmetic potential. The manuscript can be improved by improving the experimental section and more specifically the conclusion.

The specific comments, which could help to improve the manuscript are:

1.      The manuscript should be revised for grammatical & punctuation errors.

2.      Line 20: Abstract: Ceratocarpus arenarius (Chenopodiaceae) is an under-investigated desert annual with promising potential for use in the pharmaceutical, cosmetic and food industries. Correct the sentence.

3.      Present plant name always in italics “C. arenarius”. Line 23 and onwards.

4.      Line 104: The leaf trichomes of the “stellate hornbeam” have a one- or two-cell stalk followed by 104 more densely packed cells? What is stellate hornbeam?

5.      Provide GC-Ms chromatogram as supplementary material. Provide GC-MS procedure in detail. Also provide the identification of compounds by GC-MS in detail. Please refere to:

1. Analysis of fatty acid composition of Withania coagulans fruits by gas chromatography/mass spectroscopy. Research Journal of Pharmacognosy, 2017, 4(4).

2. Fatty acids analysis of Ficus religiosa stem bark by gas chromatography-mass spectrometry. International Journal of Advances in Pharmacy Medicine and Bioallied Sciences. 2017, 112.

6.      Determination of Amino Acid Composition: Provide GC chromatogram as supplementary material.

7.      Determination of Mineral Content: Provide Atomic Absorption Spectrophotometer spectra as supplementary material.

8.      Provide figures related to Cytotoxic Activity as supplementary material.

9.      Phytochemical analysis should be performed on  GC/MS or other chromatographic techniques for chemoprofiling of plant extract. It is better to perform both qualitative and quantitative analysis.

10.  Authors did not conclude their findings and future directions in an effective and significant way.

Comments on the Quality of English Language

Moderate editing of English language is required.

Author Response

The specific comments, which could help to improve the manuscript are:

  1. The manuscript should be revised for grammatical & punctuation errors

Response 1: The manuscript has been corrected for grammatical and punctuation errors by a professional native English speaker

  1. Line 20: Abstract: Ceratocarpus arenarius (Chenopodiaceae) is an under-investigated desert annual with promising potential for use in the pharmaceutical, cosmetic and food industries. Correct the sentence.

Response 2: Yes, we corrected the text.

  1. Present plant name always in italics “C. arenarius”. Line 23 and onwards.

       Response 3: Yes, we corrected the name of the plant species.

  1. Line 104: The leaf trichomes of the “stellate hornbeam” have a one- or two-cell stalk followed by 104 more densely packed cells? What is stellate hornbeam?

Response 4: Trichomes or hairs, are cells of the epidermis or outgrowths that form pubescence on the surface organs of plants, and can be present on all terrestrial organs of the plant. The hairs have a stellate shape. They can be simple and complex. In our plant, it has a complex shape. they are also called stellate hairs (trichomes). Incorrect translation. It is stellate hairs

  1. Provide GC-MS chromatogram as supplementary material. Provide GC-MS procedure in detail. Also provide the identification of compounds by GC-MS in detail. Please refere to:
  2. Analysis of fatty acid composition of Withania coagulans fruits by gas chromatography/mass spectroscopy. Research Journal of Pharmacognosy, 2017, 4(4).
  3. Fatty acids analysis of Ficus religiosa stem bark by gas chromatography-mass spectrometry. International Journal of Advances in Pharmacy Medicine and Bioallied Sciences. 2017, 112.

Response 5: Yes, we provide a chromatogram as supplementary material. And also a description of fatty acid compounds has been added

  1. Determination of Amino Acid Composition: Provide GC chromatogram as supplementary material.

Response 6: Yes, we provide a chromatogram as supplementary material.

  1. Determination of Mineral Content: Provide Atomic Absorption Spectrophotometer spectra as supplementary material.

Response 7: In our research device, the spectrum of the final analysis result is not visible, we can only see the absorption. For this reason, we cannot provide supplementary material.

  1. Provide figures related to Cytotoxic Activity as supplementary material.

Response 8: Yes, We provide figures as supplementary material.

  1. Phytochemical analysis should be performed on GC/MS or other chromatographic techniques for chemoprofiling of plant extract. It is better to perform both qualitative and quantitative analysis.

Response 9: We thank the reviewer for this comment. Since UHPLC-QTOF-MS/MS analysis of C. arenarius has already be done in the work by Le Pan et al. (UHPLC-QTOF-MS/MS based characterization of anti-tumor constituents in Ceratocarpus arenarius L. and identification of EGFR-TK inhibitors by virtual screening. Natural Product Product 2022, 36, 1-11), the focus of our study were botanical characterization, examination of proximate parameters and minerals, and cytotoxic activity.

  1. Authors did not conclude their findings and future directions in an effective and significant way.

Response 10: We modified and improved the conclusion section with a special focus on future directions.

Reviewer 2 Report

Comments and Suggestions for Authors

The authors bring information about Chenopodiaceae family and in special about Ceratocarpus arenarius L. that is a plant tolerant to high drought and salinity resistance, tolerance to nutrient deficiency and it produces around 4000 seeds.

They mention that is a pioneer plants, that can find in Kazakhstan, Central Asia. Ait has polyphenolic compounds, e.g. flavonoids and phenolic acids, organic acids, saponins, steroids, and vitamins C and B2, and it is used in folk medicine. Then, this plant may have important characteristic to be used in pharmaceutical and cosmetic industry. But it is important to know better this plant before to use in large scale. Then here, the authors make a characterization of this plant.

My recommendations

1)     Table 1 and figure 1 it was presented the plant, it can also be done a schematic representation by draw showing the phyllotaxis, flowers, seeds, etc. And in figure 1 b and 1 c can the image be improved. It is difficult to understand what they want to present,

2)     Keep the specie name in italics,

3)     Figure 4, 5 and 6 needs a scale.

4)     Table 5 is correct the value of protein and error 392.85±525.50?

The authors made an important characterization of the plant with histological analysis, made the ash composition, determination of amino acid, lipids that are important when you think to use in nutrition for example. Then they produced an ethanol extract and tested the cytotoxicity using the marine crayfish Artemia salina and showed that is not toxic. But what are the molecules in this extract? What is the other activities as the title mention cosmetic and pharmaceutical potential. There is not enough test for this title.

The manuscript has important data for this plant characterization,  and it shows a potential use, but the data present here is not for the molecules journal, it is good for example for plants journal from MPDI.

Comments on the Quality of English Language

It need to be review

Author Response

Comments and Suggestions for Authors

The authors bring information about Chenopodiaceae family and in special about Ceratocarpus arenarius L. that is a plant tolerant to high drought and salinity resistance, tolerance to nutrient deficiency and it produces around 4000 seeds.

They mention that is a pioneer plants, that can find in Kazakhstan, Central Asia. Ait has polyphenolic compounds, e.g. flavonoids and phenolic acids, organic acids, saponins, steroids, and vitamins C and B2, and it is used in folk medicine. Then, this plant may have important characteristic to be used in pharmaceutical and cosmetic industry. But it is important to know better this plant before to use in large scale. Then here, the authors make a characterization of this plant.

My recommendations

  • Table 1 and figure 1 it was presented the plant, it can also be done a schematic representation by draw showing the phyllotaxis, flowers, seeds, etc. And in figure 1 b and 1 c can the image be improved. It is difficult to understand what they want to present

Response 1:  Trichomes are numerous. Microscopic from 1-cell to 2-3-cell types. Figure 1 b and 1c show a photo that was taken with a binocular stereomicroscope with a digital magnification of 0.75 x - 3.6 x

2)     Keep the specie name in italics,

      Response 2:  Yes, in the article we corrected the name of the plant species in italics

3)     Figure 4, 5 and 6 needs a scale.

      Response 3:  In figure 4 - Magnification 10x and 40x

                              In figure 5, 6 - Magnification 40x

4)     Table 5 is correct the value of protein and error 392.85±525.50?

      Response 4:  Thank you for looking at the manuscript carefully. Due to a technical error, one digit was written unnecessarily. That's right 392.85±25.50

The authors made an important characterization of the plant with histological analysis, made the ash composition, determination of amino acid, lipids that are important when you think to use in nutrition for example. Then they produced an ethanol extract and tested the cytotoxicity using the marine crayfish Artemia salina and showed that is not toxic. But what are the molecules in this extract? What is the other activities as the title mention cosmetic and pharmaceutical potential. There is not enough test for this title.

70 % ethanolic extract has been tested for cytotoxicity

Reviewer 3 Report

Comments and Suggestions for Authors

The manuscript titled "Pharmaceutical, cosmetic and food potential of Ceratocarpus arenarius: Botanical characteristics, proximate and mineral com position, and cytotoxic activity" by Aigerim et al., describe the morphological, anatomical, nutrient, mineral contents, and cytotoxic activity of C. arenarius. This is a well-written article and I anticipate that the manuscript should be of great interest to the researchers working on plants under extreme environmental conditions. I considered the manuscript suitable for publication subject to following improvements. The abstract section should be improved by indicating the implications of the findings. Add novelty in the introduction to increase its merit for citations. The induction section is very short, required more information and some latest references from the available literature. It is suggested to revise the main objectives of the present work at the end of introduction.

A.    What specific morphological features were examined using the light microscope, and how do these features contribute to the plant's potential for pharmaceutical, cosmetic, and food applications?

B.     The title needs to revised.

C. While the presence of various minerals is mentioned, what concentrations of these minerals were found? Add significant results to the abstract section.

D.    Are there variations in the plant's composition or bioactivity in different parts of Kazakhstan?

E.     Are there any unique characteristics or structures observed in the cross sections of stems, roots, and leaves that could have practical implications?

F.  What concentrations of the ethanolic extract were tested for cytotoxicity, and how does the absence of acute toxicity against brine shrimp nauplii translate to potential safety for human use?

G. How might the geographical location, specifically Kazakhstan, influence the composition and properties of C. arenarius, and how applicable are the findings to regions with different environmental conditions?

Author Response

Comments and Suggestions for Authors

The manuscript titled "Pharmaceutical, cosmetic and food potential of Ceratocarpus arenarius: Botanical characteristics, proximate and mineral com position, and cytotoxic activity" by Aigerim et al., describe the morphological, anatomical, nutrient, mineral contents, and cytotoxic activity of C. arenarius. This is a well-written article and I anticipate that the manuscript should be of great interest to the researchers working on plants under extreme environmental conditions. I considered the manuscript suitable for publication subject to following improvements. The abstract section should be improved by indicating the implications of the findings. Add novelty in the introduction to increase its merit for citations. The induction section is very short, required more information and some latest references from the available literature. It is suggested to revise the main objectives of the present work at the end of introduction.

  1. What specific morphological features were examined using the light microscope, and how do these features contribute to the plant's potential for pharmaceutical, cosmetic, and food applications?

Response A: Morphological features that have been studied (stem, leaf and root form and dimensions, stem, leaf and root tissues: epidermis, spongy and palisade tissue, vascular tissue, collenchyma, sclerenchyma, etc.) are the basis for the botanical characterization/identification of the plant as a raw material. These characteristics are not directly related to the plant’s potential for use in pharmacy/medicine, cosmetics and as food; however, they are crucial to ensure authenticity and quality of the corresponding products.

  1. The title needs to revised.

Response C: We changed the title to “Ceratocarpus arenarius: Botanical characteristics, Proximate, Mineral Composition, and Cytotoxic Activity”.

  1. While the presence of various minerals is mentioned, what concentrations of these minerals were found? Add significant results to the abstract section.

Response C: The results are included in Table 8 and we have added concentrations of elements in the abstract

  1. Are there variations in the plant's composition or bioactivity in different parts of Kazakhstan?

Response D: Previously, this plant species had not been studied in Kazakhstan. We started the research for the first time. In the future, we want to standardize it and add it to the Pharmacopoeia of Kazakhstan

  1. Are there any unique characteristics or structures observed in the cross sections of stems, roots, and leaves that could have practical implications?

Response D: For the first time, the characteristics and structures of the cross sections of stems, roots and leaves of C. arenarius growing in Kazakhstan were studied. In general, the cross-sections of parts of plants are the same, they may differ from the size of cellular structures. Biologically active substances and nutrients accumulate in the intercellular spaces. Since macroscopic and microscopic studies of this plant have not been carried out before, our results can be used to develop a regulatory document and enter it into the pharmacopoeia of Kazakhstan

  1. What concentrations of the ethanolic extract were tested for cytotoxicity, and how does the absence of acute toxicity against brine shrimp nauplii translate to potential safety for human use?

Response F: 70 % ethanolic extract has been tested for cytotoxicity

  1. How might the geographical location, specifically Kazakhstan, influence the composition and properties of C. arenarius, and how applicable are the findings to regions with different environmental conditions?

Response G: Ecological environment of the soil, as well as weather conditions,directly affect the plant metabolism and thus the composition and properties of the extracts. Since the research on C. arenarius is very scarce, no data are available on this topic and this will be the focus of our future research.

Round 2

Reviewer 1 Report

Comments and Suggestions for Authors

The authors have justified the comments except:

Comment 4: Still “stellate hornbeam” is mentioned in line 115.

Comment 5. Provide figure legend in supplementary material. Still procedure of GC/MS analysis and Identification of chemical components are not presented in detail. Please refer to GC/MS analysis, and Identification of components heading in:

Analysis of fatty acid composition of Withania coagulans fruits by gas chromatography/mass spectroscopy. Research Journal of Pharmacognosy, 2017, 4(4).

Comment 6. Provide figure legend in supplementary material.

Comment 8. The supplementary figures provided are irrelevant. Provide figures related to Cytotoxic Activity showing dead or viable A. salina.

Refer to the: Toxicity Assessment of Lactococcus lactis IO-1 Used in Coconut Beverages against Artemia salina using Brine Shrimp Lethality Test. Applied Food Biotechnology, 2020, 7(3), 127–134. https://doi.org/10.22037/afb.v7i3.29346

Line 383: Revised the sentence “The marine crayfish Artemia salina were taken to determine cytotoxic activity”. Cray fish and Artemia salina are not identical species. 

Comments on the Quality of English Language

Minor editing of English language is required.

Author Response

Comments and Suggestions for Authors

The authors have justified the comments except:

Comment 4: Still “stellate hornbeam” is mentioned in line 115.

Response 4: We corrected "stellate hornbeam" to "stellate hairs" in the manuscript and highlighted it in yellow

Comment 5. Provide figure legend in supplementary material. Still procedure of GC/MS analysis and Identification of chemical components are not presented in detail. Please refer to GC/MS analysis, and Identification of components heading in:

Analysis of fatty acid composition of Withania coagulans fruits by gas chromatography/mass spectroscopy. Research Journal of Pharmacognosy, 2017, 4(4).

Response 5: We added procedure of GC/MS analysis and Identification to the methodology

Comment 6. Provide figure legend in supplementary material.

Response 6: We thank the reviewer for this comment. We have indicated figure legend in supplementary material and are sending them again

Comment 8. The supplementary figures provided are irrelevant. Provide figures related to Cytotoxic Activity showing dead or viable A. salina.

Refer to the: Toxicity Assessment of Lactococcus lactis IO-1 Used in Coconut Beverages against Artemia salina using Brine Shrimp Lethality Test. Applied Food Biotechnology, 2020, 7(3), 127–134. https://doi.org/10.22037/afb.v7i3.29346

Response 8: We thank the reviewer for this comment. The number of dead and viable A. salina was examined under a microscope and calculated. We didn't make figures. For this reason, we cannot provide supplementary material

Line 383: Revised the sentence “The marine crayfish Artemia salina were taken to determine cytotoxic activity”. Cray fish and Artemia salina are not identical species. 

There was an error in translation. There should be Artemia salina everywhere. We've corrected it

Reviewer 3 Report

Comments and Suggestions for Authors

Accept in the present form

Author Response

Accept in the present form